# Enamel Matrix Derivative Decreases Pyroptosis-Related Genes in Macrophages

**DOI:** 10.3390/ijms23095078

**Published:** 2022-05-03

**Authors:** Mariane Beatriz Sordi, Ariadne Cristiane Cabral da Cruz, Layla Panahipour, Reinhard Gruber

**Affiliations:** 1Department of Oral Biology, University Clinic of Dentistry, Medical University of Vienna, 1090 Vienna, Austria; mariane.sordi@kcl.ac.uk (M.B.S.); layla.panahipour@meduniwien.ac.at (L.P.); 2Department of Dentistry, Federal University of Santa Catarina, Florianopolis 88040-900, Brazil; ariadne.cruz@ufsc.br; 3Department of Periodontology, School of Dental Medicine, University of Bern, 3010 Bern, Switzerland

**Keywords:** enamel matrix derivative, pyroptosis, inflammasomes, periodontal diseases, macrophages, mesenchymal cells

## Abstract

**Background**: Pyroptosis is a caspase-dependent catabolic process relevant to periodontal disorders for which inflammation is central to the pathophysiology of the disease. Although enamel matrix derivative (EMD) has been applied to support periodontal regeneration, its capacity to modulate the expression of pyroptosis-related genes remains unknown. Considering EMD has anti-inflammatory properties and pyroptosis is linked to the activation of the inflammasome in chronic periodontitis, the question arises whether EMD could reduce pyroptosis signalling. **Methods**: To answer this question, primary macrophages obtained from murine bone marrow and RAW 264.7 macrophages were primed with EMD before being challenged by lipopolysaccharide (LPS). Cells were then analysed for pyroptosis-signalling components by gene expression analyses, interleukin-1β (IL-1β) immunoassay, and the detection of caspase-1 (CAS1). The release of mitochondrial reactive oxygen species (ROS) was also detected. **Results**: We report here that EMD, like the inflammasome (NLRP3) and CAS1 specific inhibitors—MCC950 and Ac-YVAD-cmk, respectively—lowered the LPS-induced expression of NLRP3 in primary macrophages (EMD: *p* = 0.0232; MCC950: *p* = 0.0426; Ac-YVAD-cmk: *p* = 0.0317). EMD further reduced the LPS-induced expression of NLRP3 in RAW 264.7 cells (*p* = 0.0043). There was also a reduction in CAS1 and IL-1β in RAW 264.7 macrophages on the transcriptional level (*p* = 0.0598; *p* = 0.0283; respectively), in IL-1β protein release (*p* = 0.0313), and CAS1 activity. Consistently, EMD, like MCC950 and Ac-YVAD-cmk, diminished the ROS release in activated RAW 264.7 cells. In ST2 murine mesenchymal cells, EMD could not be tested because LPS, saliva, and IL-1β + TNF-α failed to provoke pyroptosis signalling. **Conclusion**: These findings suggest that EMD is capable of dampening the expression of pyroptosis-related genes in macrophages.

## 1. Introduction

Periodontal disease is a global health problem [1]. Currently, peri-implant disease has reached the same level of concern as periodontal disorders. In this scenario, mucointegration—the attachment of soft tissues to the transmucosal portion of an implant—is just as relevant for implant success as osseointegration [2]. A disruption in mucointegration can manifest as peri-implant mucositis and, if not resolved, can progress to inflammatory peri-implantitis [3,4]. Periodontitis and peri-implantitis are universally agreed to begin with a breakdown in the soft tissue attachment and bone loss progression [5,6]. Consequently, methods to strengthen, maintain, or regenerate the soft tissue attachment around the tooth or the dental implant are critical for improving the protection sealing against microbial infections or endogenous danger signals [7]. The underlying pathogenesis of periodontitis/peri-implantitis is a chronic inflammation that drives downstream catabolic cellular events ultimately leading to tooth loss due to a lack of supporting tissues [6,8,9]. There is thus a critical requirement to understand the fundamental pathological mechanisms on a cellular and molecular basis to implement therapies aiming to regulate inflammation and thereby pave the way for regenerative strategies [8,9]. Thus, understanding the pathways connecting inflammation and tissue destruction will help to develop strategies to prevent and treat periodontitis and peri-implantitis.

Pyroptosis is an inflammatory caspase-dependent catabolic process that is relevant for innate immunity. This process is mainly mediated by the activation of caspase-1 (CAS1) by the nucleotide-binding domain (NBD) and leucine-rich repeat (LRR)-containing protein 3 (NLRP3) inflammasome [10]. Then, CAS1 cleaves the gasdermin D (GSDMD), which is responsible for cell membrane perforation and the release of interleukins-1β (IL-1β) and -18 (IL-18) [10], which, in turn, trigger a robust inflammatory response on the surrounding tissues [11]. NLRP3 and CAS1 are important for bacterial clearance; however, if overexpressed, they may lead to cellular self-destruction, inflammation, and tissue damage [12]. Immunostaining images showed a stronger signalling intensity for NLRP3, cleaved CAS1, and IL-1β in the connective tissue of periodontitis compared to a healthy gingiva [13]. Additionally, using a periodontitis mouse model, higher amounts of NLRP3 and IL-1β were visible in the inflamed gingiva [13]. There is thus evidence for pyroptosis to occur in periodontal diseased tissues.

In vitro periodontal models in pyroptosis research focus on the NLRP3/CAS1/GSDMD-mediated pyroptosis pathway in monocytes, macrophages, and periodontal ligament cells [11,14,15,16]. NLRP3 inflammasome can react to a wide range of bacterial ligands and play a pivotal role in the pathogenesis of inflammatory diseases. Lipopolysaccharide (LPS) is a virulence factor and a strong agonist of toll-like receptor (TLR) that is able to initiate pyroptosis signalling [16,17]. LPS is produced by Gram-negative bacteria [18]. Considering that oral diseases are mainly mediated by Gram-negative bacteria, it makes sense that LPS is related to periodontal disorders [12,18,19]. Taking advantage of this in vitro model, glycogen synthase kinase-3β (GSK-3β) deficiency was identified to lower the LPS-induced pyroptosis through the inactivation of NLRP3 inflammasome [16]. Accordingly, NLRP3/CAS1/GSDMD-mediated pyroptosis bioassays are suitable for identifying the components that lower pyroptosis signalling. Furthermore, considering the impairment caused by pyroptosis on periodontal disorders, finding ways to inhibit or reduce pyroptosis downstream brings prospects for periodontal therapies.

Enamel matrix derivative (EMD) is a xenograft applied to support periodontal regeneration [20] that was also considered a treatment for venous leg ulcers [21]. EMD is an extract of enamel matrix from the tooth germ of piglets and propylene glycol alginate serves as a matrix. Proteome analyses confirmed the presence of enamel matrix proteins amelogenin and ameloblastin [22], and growth factors such as TGF-β have also been identified [23,24]. More importantly for this paper, EMD has been shown to exert anti-inflammatory activity in vitro. LPS-stimulated rat monocytes exposed to EMD exhibited a decrease in TNF-α production [25]. In human blood-derived cells exposed to LPS and peptidoglycan, EMD lowered TNF-α release [26]. In LPS-stimulated human osteogenic cells and immortalized human epithelial gingival keratinocytes, EMD lowered the expression of inflammatory cytokines including TNF-α [27]. Nevertheless, the expression of pyroptosis factors in cells stimulated with pyroptosis-triggering dangers—and primed with EMD—has not yet been explored. It might be hypothesized that the beneficial effects of EMD [25,26,27] are caused by lowering the pyroptosis-mediated cellular self-destruction and inflammation in periodontitis.

Since there is strong in vitro evidence that EMD has anti-inflammatory properties [25,26,27] and pyroptosis is linked to the activation of the inflammasome in chronic periodontitis and peri-implantitis [11,12,16], the question arises whether EMD could reduce pyroptosis in vitro. Therefore, we tested the hypothesis that the anti-inflammatory activity of EMD is at least partially involved in a lowering of the LPS-mediated pyroptosis factors.

## 2. Materials and Methods

### 2.1. Primary Macrophages, RAW 264.7 Macrophage-like Cells, and ST2 Mesenchymal Cells

BALB/c mice of 6 to 8 weeks old were purchased from Animal Research Laboratories, Himberg, Austria. Bone marrow cells were collected from the femora and tibiae as previously described [28]. Briefly, mice were sacrificed, and the femora and tibiae were removed. Bone marrow cells were seeded at 1 × 10^6^ cells/cm^2^ into 24-well plates and grown for 7 days in Dulbecco’s Modified Essential Medium (DMEM; Sigma Aldrich, St. Louis, MO, USA) supplemented with 10% fetal calf serum (FCS; Capricorn Scientific GmbH, Ebsdorfergrund, Germany), 1% antibiotics (PS; Sigma Aldrich, St. Louis, MO, USA), and 20 ng/mL macrophage colony-stimulating factor (M-CSF; ProSpec, Ness-Ziona, Israel). RAW 264.7 macrophage-like cells (LGC Standards, Wesel, Germany) were expanded in growth medium and seeded at 3 × 10^5^ cells/cm^2^ into 24-well plates. ST2 murine mesenchymal cells (Riken Cell Bank, Tsukuba, Japan) isolated from mouse bone marrow were seeded at 3 × 10^5^ cells/cm^2^ into 24-well plates. Cells were primed with 30 µg/mL of enamel derivative matrix (EMD; Straumann AG, Switzerland) for 1 h and then exposed to 100 ng/mL of LPS from *Escherichia coli* 055:B5 (Sigma Aldrich, St. Louis, MO, USA) for 6 h to induce an inflammatory response. Alternatively, 5% saliva [29] or 20 ng/mL IL-1β (ProSpec, Ness-Ziona, Israel) and TNF-α (ProSpec, Ness-Ziona, Israel) were used for cell stimulation. Pyroptosis-specific inhibitors were applied to establish the in vitro LPS-induced pyroptosis model: MCC950 (CP-456773 Sodium, Selleck Chemicals GmbH, Houston, TX, USA) was applied at 8 µM for 30 min before cells were exposed to LPS and Ac-YVAD-cmk (≥95%, HPLC; Sigma Aldrich, St. Louis, MO, USA) was applied at 5 µM for 20 h prior to the LPS challenge. All cell lineages were exposed to the respective treatments under standard conditions of 37 ◦C, 5% CO_2_, and 95% humidity.

### 2.2. Reverse Transcription Quantitative Real-Time PCR (RT-qPCR) and Immunoassay

For RT-qPCR, after stimulation, total RNA was isolated with the ExtractMe total RNA kit (Blirt S.A., Gdańsk, Poland) followed by reverse transcription and polymerase chain reaction (LabQ, Labconsulting, Vienna, Austria) on a CFX Connect Real-Time PCR Detection System (Bio-Rad Laboratories, Hercules, CA, USA). The mRNA levels were calculated by normalizing to the housekeeping gene GAPDH using the ^ΔΔ^Ct method.

The primer sequences were mNLRP3-F: 5′-TCACAACTCGCCCAAGGAGGAA-3′; mNLRP3-R: 5′-AAGAGACCACGGCAGAAGCTAG-3′; mCAS1-F: 5′-GGCACATTTCCAGGACTGACTG-3′; mCAS1-R: 5′-GCAAGACGTGTACGAGTGGTTG-3′; mCAS11-F: 5′-CCTGAAGAGTTCACAAGGCTT-3′; mCAS11-R: 5′-CCTTTCGTGTAGGGCCATTG-3′; mGSDMD-F: 5′-GGTGCTTGACTCTGGAGAACTG-3′; mGSDMD-R: 5′-GCTGCTTTGACAGCACCGTTGT-3′; mIL-1β-F: 5′-CAACCAACAAGTGATATTCTCCATG-3′; mIL-1β-R: 5′-GATCCACACTCTCCAGCTGCA-3′; mIL-18-F: 5′-CAAACCTTCCAAATCACTTCCT-3′; mIL-18-R: 5′-TCCTTGAAGTTGACGCAAGA-3′; mGAPDH-F: 5′-AACTTTGGCATTGTGGAAGG-3′; mGAPDH-R: 5′-GGATGCAGGGATGATGTTCT-3′. RT-PCR data are represented compared to the untreated control. Supernatants and the respective cell lysates prepared with 0.3% Triton X-100 (Sigma Aldrich, St. Louis, MO, USA) were analyzed for IL-1β secretion by immunoassay (R&D Systems, Minneapolis, MN, USA) according to the manufacturer’s instruction.

### 2.3. Western Blot

RAW 264.7 cells were seeded at 3 × 10^5^ cells/cm^2^ into 12-well plates. On the following day, serum-starved cells were primed with EMD for 1 h and then exposed to LPS for another 6 h. Extracts containing SDS buffer with protease and phosphatase inhibitors were separated by SDS-PAGE (cOmplete ULTRA Tablets and PhosSTOP; Roche, Mannheim, Germany) and transferred onto PVDF membranes (Roche Diagnostics, Mannheim, Germany). Membranes were blocked and the binding of the Caspase-1 (D7F10), gasdermin D (E8G3F), and cleaved gasdermin D (E7H9G) first antibodies (rabbit IgG, 1:1000; Cell Signaling Technology, Danvers, MA, USA) were detected with the second antibody labelled with HRP (goat anti-rabbit IgG, 1:10,000; Cell Signaling Technology, Danvers, MA, USA). After exposure to the Clarity Western ECL Substrate (Bio-Rad Laboratories Inc., Hercules, CA, USA) chemiluminescence signals were visualized with the ChemiDoc imaging system (Bio-Rad Laboratories Inc., Hercules, CA, USA). Quantification of band intensity was performed using ImageJ software.

### 2.4. Mitochondrial Reactive Oxygen Species (ROS) Release

RAW 264.7 cells were seeded at 3 × 10^5^ cells/cm^2^ into 96-well plates and followed the standard stimulation with EMD, MCC950, or Ac-YVAD-cmk, then challenged with LPS for 6 h. Cells were analysed for the release of mitochondrial reactive oxygen species (MitoROS 580, AAT Bioquest, Inc., Sunnyvale, CA, USA) according to the manufacturer’s instructions.

### 2.5. Statistical Analysis

All experiments were performed at least three times. Statistical analyses of gene expression and immunoassays were performed with paired t-tests, while ROS release statistical analyses were performed with one-way ANOVA followed by Dunnett’s multiple comparison test. Analyses were performed using Prism v.9 (GraphPad Software, La Jolla, CA, USA). Significance was set at *p*  < 0.05.

## 3. Results

### 3.1. Pyroptosis Inhibitors Validate Macrophages to Serve as a Pyroptosis Model

To establish a pyroptosis model, primary macrophages generated from murine bone marrow were exposed to *E. coli* LPS. MCC950 and Ac-YVAD-cmk were introduced as inhibitors raised against NLRP3 and CAS1, respectively. MCC950 significantly reduced the forced expression of NLRP3, CAS11, and IL-1β, but also consistently decreased CAS1 and IL-18 gene expressions in primary macrophages. Likewise, Ac-YVAD-cmk significantly reduced the expression of NLRP3 and IL-18, and showed a trend to the reduction in the expression of CAS1, CAS11, and IL-1β, in primary macrophages (Figure 1). These findings support the LPS-induced primary macrophages to serve as a bioassay to test EMD and its potential for reducing pyroptosis signalling.

### 3.2. EMD Reduces the Expression of Pyroptosis Markers in LPS-Induced Primary Macrophages

To test EMD and its potential for reducing pyroptosis in the established bioassay, primary macrophages were primed with EMD before being challenged by LPS and then were analysed for gene expression of pyroptosis signalling components. Our chosen dose of 30 µg/mL EMD did not lead to any cytotoxicity either alone or in combination with LPS (data not shown); therefore, we proceeded with the gene expression analyses. LPS caused a robust increase in the expression of the pyroptosis genes NLRP3, CAS1, CAS11, IL-1β, and IL-18 in primary macrophages, with particularly strong expressions of NLRP3 and IL-1β. EMD significantly lowered the LPS-induced expression of NLRP3, CAS1, and IL-18, suggesting that primary macrophages are susceptible to EMD and its pyroptosis-lowering activity (Figure 2).

### 3.3. EMD Reduces the Expression of Pyroptosis Markers in LPS-Induced RAW 264.7 Macrophages

To implement a cell line-based pyroptosis model, RAW 264.7 macrophages were exposed to LPS followed by the screening for the respective pyroptosis marker genes. Consistent with the findings regarding the primary macrophages, EMD significantly reduced the LPS-induced expression of NLRP3 and IL-1β. There was also a trend toward reducing CAS1 and CAS11 expression (Figure 3).

Differently from primary macrophages though, it was mainly the IL-1β but not the IL-18 expression that was reduced by EMD in RAW 264.7 cells. As expected [30], immunoassays of RAW 264.7 cells revealed negligible amounts of IL-1β in the supernatant (Appendix A). Nevertheless, under the permeabilization of the cell membrane, IL-1β could be confirmed in LPS-stimulated RAW 264.7 cells as well as the significant IL-1β reduction with the treatment with EMD (Figure 4A).

Moreover, EMD reduced cleaved CAS1 at the protein level (Figure 4B), suggesting a decrease in the CAS1 activity and that EMD could lower the expression and the activation of CAS1 by NLRP3 reduction. The bands were quantified regarding intensity (Appendix A), confirming what can be pictured in the Western blot images. Thus, the RAW 264.7 macrophages are suitable to identify EMD for lowering a pyroptosis response.

### 3.4. EMD Reduces Reactive Oxygen Species (ROS) in LPS-Induced RAW 264.7 Macrophages

RAW 264.7 macrophages were again exposed to LPS and analysed for mitochondrial ROS release. EMD reduced the mitochondrial ROS release in RAW 264.7 cells to levels comparable to the untreated control, suggesting a reduction in cellular stress levels by the EMD treatment. Consistently, the pyroptosis specific inhibitors, MCC950 and Ac-YVAD-cmk, diminished ROS release in activated RAW 264.7 cells (Figure 5).

### 3.5. ST2 Mesenchymal Cells Are Not Suitable to Test the Potential Role of EMD on Pyroptosis

Finally, we introduced LPS and saliva stimulation over ST2 murine mesenchymal cells to serve as a model for pyroptosis testing. However, both, LPS and saliva, failed to considerably increase the expression of the most sensitive pyroptosis marker—the NLRP3—and all other pyroptosis markers, including IL-1β and IL-18, suggesting that neither LPS nor saliva stimulation in ST2 cells were suitable models to evaluate EMD to change pyroptosis (Appendix A). When ST2 cells were exposed to IL-1β and TNF-α, there was a strong increase of interleukin-6 (IL-6) and chemokines CCL2 and CXCL2, which were further reduced by EMD application (Figure 6). Nevertheless, no changes in NLRP3 or any other pyroptosis markers were found (Appendix A). Thus, LPS, saliva, or IL-1β + TNF-α stimulations on ST2 cells are not applicable to evaluate the potential role of EMD to reduce pyroptosis signalling.

## 4. Discussion

Pyroptosis is a major driver of inflammatory disorders and is chiefly activated by NLRP3 inflammasome and caspases. Thus, NLRP3 and CAS1, the hallmarks of pyroptosis signalling, are increasingly expressed in periodontal disease compared to healthy tissue [10,12,31]. Considering EMD is widely used in periodontal regeneration and has demonstrated anti-inflammatory properties in vitro [20,25,26,27], we hypothesized that part of the beneficial activity of EMD might involve the modulation of pyroptosis signalling. Indeed, our major finding was that EMD lowered the forced expression of NLRP3 and CAS1 activity in murine macrophage models. Taken together, our findings suggest that EMD diminishes pyroptosis signalling in macrophages.

If we relate our findings to those of other studies, our data completes the overall picture of the anti-inflammatory activity that EMD has in vitro on various models from human, rat, and mouse cells [25,26,27,32]. However, these models mainly used TNFα or interferon-gamma (IFNγ) to simulate inflammation, but TNFα and IFNγ are not drivers of pyroptosis signalling. It was merely the study on LPS-stimulated human osteoblastic cells and human gingival keratinocytes that assessed EMD lowering the expression of IL-1β, however, this study was not focused on pyroptosis [27]. Hence, our findings that EMD reduces the expression of IL-1β in RAW 264.7 macrophages support the existing knowledge on the anti-inflammatory properties of EMD; nevertheless, this observation is not sufficient to support the involvement of EMD in the reduction of the pyroptosis signalling.

Macrophages can be polarized into either classically activated pro-inflammatory (M1) or alternatively activated anti-inflammatory (M2) macrophages depending on the stimulation [33]. M1 macrophages are induced by pathogen-associated molecular patterns (PAMPs), such as the bacterial LPS used herein, or Th1 cytokines such as IFNγ, producing a wide range of cytokines, such as TNF-α, IL-1β, IL-6, and inducible nitric oxide synthase (iNOS), to aggravate inflammation. In contrast, M2 macrophages are induced by Th2 cytokines such as IL-4 and IL-13, and they possess the ability to express arginase-1 (Arg1), chitinase-like 3 (or Ym1), and IL-10 to promote reparative processes and relieve inflammation [34]. Therefore, since we have applied LPS, we know that we are working with M1 pro-inflammatory macrophages. Furthermore, since this is a pyroptosis-related article, we did not focus on IFNγ, TNF-α, IL-6, or iNOS, but on IL-1β and IL-18, which are directly related to pyroptosis. Moreover, the production of ROS is a hallmark of M1 macrophages, which also contributes to the M2 polarization switch [35]. Nevertheless, since we have scientific support that we are working with M1 macrophages, we can affirm that the release of ROS is related to the pro-inflammatory aspect.

Our data showing that EMD significantly reduced the expression of IL-18 in primary macrophages and that NLRP3 and CAS1 specific inhibitors (MCC950 and Ac-YVAD-cmk, respectively) exert a similar activity, can be considered indirect support for EMD to attenuate pyroptosis activity. These findings are in line with other observations showing that MCC950 inhibited IL-18 release in THP1 and monocytes [36,37], reversed the forced IL-1β and IL-18 expression on periodontal ligament fibroblasts [38], HCT116 colorectal cells [39], and canine kidney epithelial cells [40]. Furthermore, Ac-YVAD-cmk reduced the forced expression of IL-18 in whole blood cells [41], in THP-1 cells [42], and also in sepsis-induced acute kidney injury [43]. Even though EMD performs similarly to MCC950 and Ac-YVAD-cmk inhibitors and reduces the expression of NLRP3, CAS1, and IL-18 in primary macrophages, this is not sufficient evidence that EMD reduces pyroptosis activity and should be supported by additional investigation.

Support for EMD to regulate pyroptosis arises from findings that EMD reduces the LPS-induced expression of NLRP3 and IL-1β in RAW 264.7 macrophages. Considering that NLRP3 together with IL-1β and IL-18 are NF-kB-target genes, it can be hypothesized that EMD lowers the LPS-driven NF-kB signalling pathway and thereby the transcription of NLRP3 and IL-1β/IL-18. Consequently, the assembly of the inflammasome is limited by the accessibility of the reduced NLRP3, and our observation that EMD lowers the LPS-induced CAS1 activity supports this concept. Thus, our findings add to the existing knowledge of the anti-inflammatory properties of EMD and guide it towards the regulation of the pyroptosis pathway in macrophages. Furthermore, our data on EMD reducing inflammation in ST2-challenged cells also give additional support to the anti-inflammatory activity of EMD in vitro.

Consistent with other reports [30], immunoassays failed to detect IL-1β in the extracellular media in LPS-stimulated RAW 264.7 cells, while the cell membrane permeabilization allowed the detection of IL-1β in LPS-challenged cells. This seems to be related to the weak GSDMD activity that is herein reported. GSDMD is required for IL-1β release in pyroptotic cells or hyperactivated cells [30]. GSDMD knockout cells are unable to form pores and release IL-1β or lactate dehydrogenase (LDH), a molecule that shows signs of membrane pore formation [30]. This agrees with our finding that LDH release was not substantially increased in LPS-stimulated RAW 264.7 macrophages (Appendix A). Likewise, GSDMD is necessary for the release of cleaved IL-1β during infection but is not required for IL-1β processing within cells [30]. Hence, it seems that our model failed to cause membrane pore formation due to reduced GSDMD activity. Thus, our model is valid to test for pyroptosis signalling but not for full pyroptosis induction, including membrane disintegration.

Regarding ROS release, EMD in LPS-stimulated RAW 264.7 macrophages reduced mitochondrial ROS, such as the NLRP3 and CAS1 specific inhibitors (MCC950 and Ac-YVAD-cmk, respectively). In agreement with our findings, MCC950 inhibited the excessive production of ROS in chondrocytes [44], and Ac-YVAD-cmk blocked the forced ROS production in HT22 cells [45] and cerebellar granule neurons [46]. Increased ROS levels drive the transcription nuclear factor and induce the pyroptosis of nucleus pulposus cells through the NLRP3 pathway which is related to the mechanism of degenerative disorders [47]. More importantly, ROS acts downstream of gene transcription, mRNA translation, and IL-1β converting enzyme activation [46]. Furthermore, ROS production occurs after K^+^ deprivation [30,46], which can induce pyroptosis [12]. Therefore, evaluating ROS release is relevant to pyroptosis signalling as part of the downstream events occurring in the pyroptotic cells.

The clinical relevance of our findings remains at the level of speculation. Clinically, EMD stabilizes blood clots and improves clinical healing in deep pockets after non-surgical periodontal treatment [48]. Minimally invasive periodontal surgery with EMD in periodontitis-affected subjects results in lower values of C-reactive protein as no inflammatory perturbation was noticed [49]. EMD treatment also reduced bleeding on probing and periodontal pocket depth, and post-surgical gingival recession was lowered [49]. Moreover, EMD shows an antibacterial effect on the viability of ex vivo supragingival dental plaque flora collected from patients with periodontitis [50]. Considering that EMD lowers the inflammation also in vivo [51] and that periodontitis is linked to pyroptosis [11,12,16], we can speculate that EMD exerts its beneficial effect by reducing pyroptosis signalling at sites of chronic periodontitis, likely involving the NLRP3 expression.

The complexity of the in vivo situation, however, is not fully represented by primary macrophages or cell lines. Primary macrophages are closer to the in vivo situation than cell lines and, therefore, we have used the primary macrophages to establish the pyroptosis system and to perform the proof-of-principle experiments. However, to reduce and replace animal organ donation—in this case, bone marrow—once we had established our model with the use of pyroptosis-specific inhibitors and evidence that EMD reduces pyroptosis-related genes, we switched to a macrophage cell line. As expected, the cell line performed similarly, although not identically, to the primary cell line. Furthermore, in vitro models are useful for identifying potential cellular responses and signalling pathways that can later be evaluated in a complex in vivo environment. By showing that EMD lowers the LPS-induced expression of pyroptosis-related genes, we provide a fundament for future research in this direction.

This study has the limitations of the in vitro research. For instance, which and how EMD component molecules responsible for the anti-pyroptosis activity reach the target cells in vivo were not explored. Since we have not discovered the molecular structure and the characteristics of the anti-inflammatory components of EMD, the in vitro findings cannot easily be translated into a clinical perspective. Hence, further studies of EMD in the inhibition of pyroptosis in periodontal tissues should be conducted in vivo. Since EMD is available for clinical purposes, studies on its impact on the periodontium are feasible. Another limitation of our model is that LPS alone was not sufficient to increase the expression of, or activate, GSDMD, an executor of pyroptosis required for the IL-1β secretion in macrophages [52]. Future studies could therefore include pyroptosis agonists such as α-hemolysin [53,54], nigericin [30], or ATP [55], together with LPS to impulse cytotoxicity and IL-1β secretion other than the gene expression of pyroptosis-related factors, i.e., the full picture of pyroptosis [52]. It might also be worth considering the impact of EMD on the CAS3 dependent apoptotic pathway, downstream of CAS1 and independent of GSDMD [56]. Further proof for EMD to reduce pyroptosis-mediated periodontal destruction might be based on mouse models with a genetic deletion of NLRP3, CAS1, or GSDMD; hypothetically, EMD cannot exert its beneficial activity when pyroptosis is blocked at the genetic level.

In conclusion, our findings suggest that EMD is capable of dampening pyroptosis-related genes in macrophages. This is relevant as the clinical use of EMD in periodontal therapies could comprise the reduction of pyroptosis downstream under conditions of periodontal tissue inflammation.

## Figures and Tables

**Figure 1 ijms-23-05078-f001:**
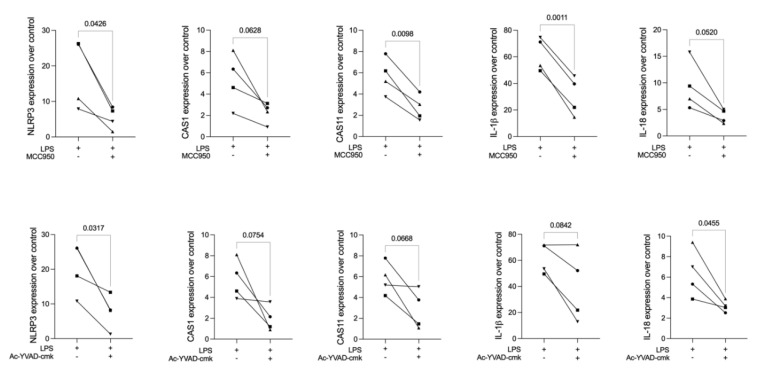
LPS stimulation caused an increase in the expression of the pyroptosis genes NLRP3, CAS1, CAS11, IL-1β, and IL-18 in primary macrophages. The application of MCC950 prior to LPS stimulation in primary macrophages led to a significant reduction in the expression of NLRP3, CAS11, and IL-1β, and a trend in the reduction of CAS1 and IL-18. The application of Ac-YVAD-cmk prior to LPS stimulation in primary macrophages led to a significant reduction in the forced expression of NLRP3 and IL-18, and a trend in the reduction of CAS1, CAS11, and IL-1β. Different symbol shapes mean independent experiments. Paired t-test statistical analysis was applied to compare the groups.

**Figure 2 ijms-23-05078-f002:**
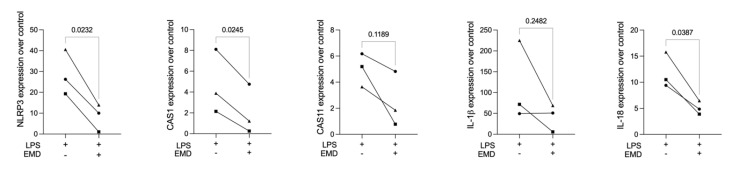
LPS stimulation caused an increase in the expression of the pyroptosis genes in primary macrophages. The application of EMD prior to LPS stimulation in primary macrophages led to a reduction in the forced expression of NLRP3, CAS1, and IL-18, and a trend in the reduction of CAS11 and IL-1β in primary macrophages. Different symbol shapes mean independent experiments. Paired *t*-test statistical analysis was applied to compare the groups.

**Figure 3 ijms-23-05078-f003:**
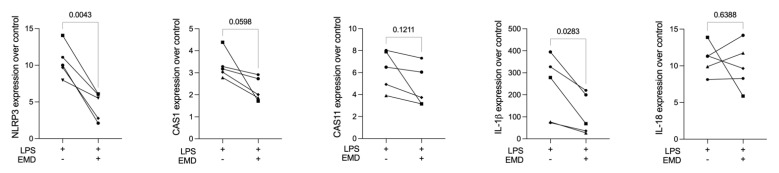
LPS stimulation caused an increase in the expression of the pyroptosis genes in RAW 264.7 macrophages. The application of EMD prior to LPS stimulation in RAW 264.7 cells led to a significant reduction in the forced expression of NLRP3 and IL-1β, and a trend in the reduction of CAS1 and CAS11 gene expression. Different symbol shapes mean independent experiments. Paired *t*-test statistical analysis was applied to compare the groups.

**Figure 4 ijms-23-05078-f004:**
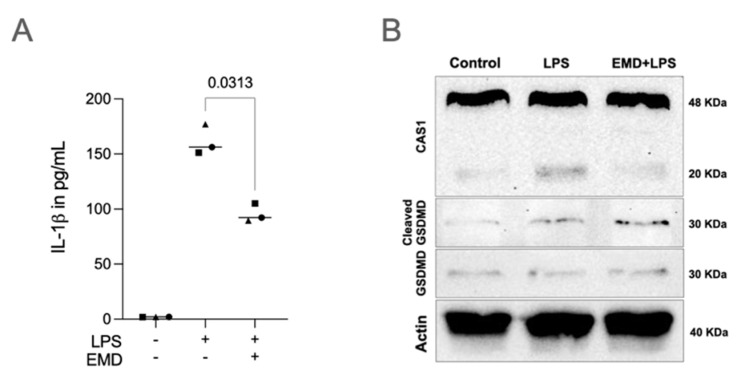
EMD reduces the pyroptosis factors in LPS-induced RAW 264.7 macrophages. (**A**) EMD protection on LPS-stimulated RAW 264.7 macrophages led to IL-1β reduction detected from the immunoassay. Different symbol shapes mean independent experiments. Paired *t*-test to compare LPS and EMD + LPS groups was applied. (**B**) Confirming the gene expression, Western blot analyses showed less cleaved CAS1 (20 KDa) protein expression in RAW 264.7 cells primed with EMD. Cleaved GSDMD was present in cells stimulated with LPS and GSDMD was present in all groups.

**Figure 5 ijms-23-05078-f005:**
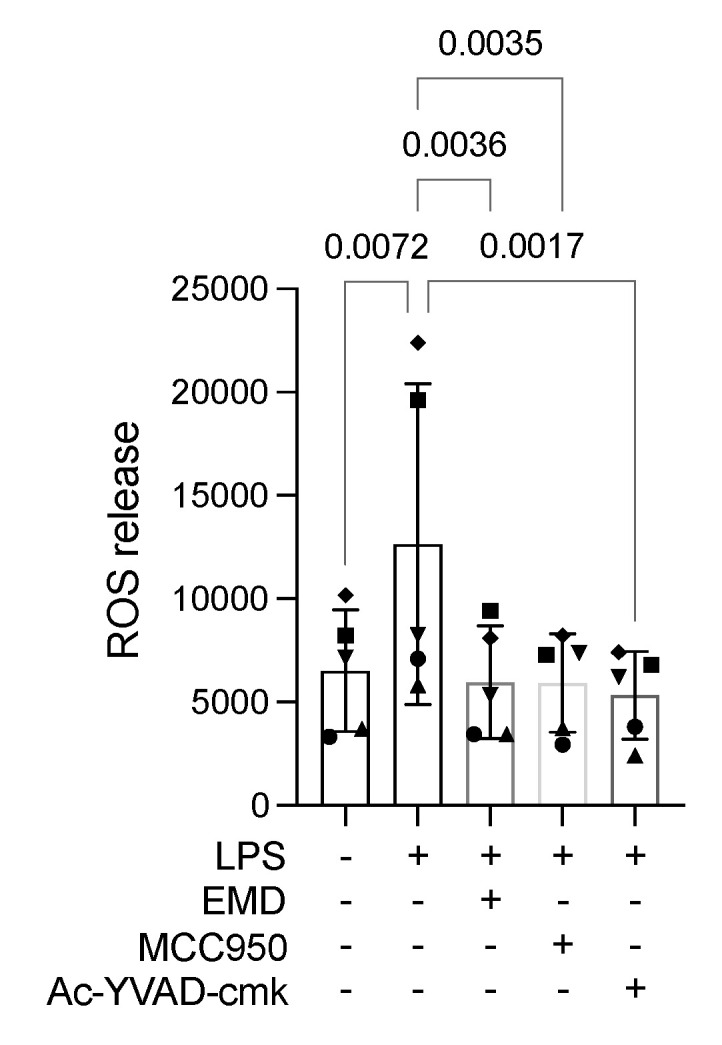
LPS stimulation caused an increase in the reactive oxygen species (ROS) release in RAW 264.7 macrophages. The application of EMD, MCC950, or Ac-YVAD-cmk in LPS-induced RAW 264.7 cells showed a significant reduction in ROS release. Different symbol shapes mean independent experiments. Repeated measures of one-way ANOVA followed by Dunnett’s multiple comparison tests, comparing every group to the LPS group, were applied.

**Figure 6 ijms-23-05078-f006:**
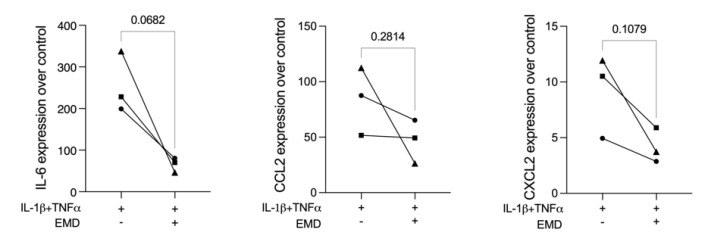
IL-1β and TNF-α stimulation caused a strong expression of IL-6, CCL2, and CXCL2 in ST2 cells. The application of EMD prior to IL-1β + TNF-α stimulation in ST2 cells led to a trend in the reduction of the forced expression of inflammatory markers. Different symbol shapes mean independent experiments. Paired *t*-test statistical analysis was applied to compare the groups.

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
