# Peer review of "Enamel Matrix Derivative Decreases Pyroptosis-Related Genes in Macrophages"

_ijms, 2022, doi:10.3390/ijms23095078_

Round 1

Reviewer 1 Report

In the present manuscript entitled "Enamel Matrix Derivative Inhibits Pyroptosis in Macrophages", the authors address the effect of Enamel Matrix Derivative (EMD) on inflammatory gene expression in macrophages. This unfortunately does nothing to address the effect of EMD on pyroptosis. The authors seem to lack an understanding of inflammasomes and pyroptosis, as they interpret a decrease of gene expression of certain inflammasome components as a readout for pyroptosis. I therefore have to recommend rejection of the manuscript, as non of the experiments presented in the manuscript actually shows what the title claims. The only thing shown is that EMD somehow decreases inflammatory gene expression.

Further points: MCC950 and YVAD are NLRP3 and Caspase 1 inhibitors respectively. They inhibit downstream outcomes of inflammasome activation, not gene expression. It is not clear to me how the authors get this effects on gene expression with their "positive controls"

Statistics shown often don't compare the relevant groups, i.e. in 4A LPS only vs LPS+EMD should be tested

It is completely unclear what one is supposed to see 4C.

Reviewer 2 Report

In this in vitro study, enamel matrix derivates (EMS) were used to modulate pyroptosis in primary mouse macrophages and a macrophage cell line. Authors show that several parameters of pyroptosis were reduced in both model systems and conclude, that EMD might be capable of dampening pyroptosis in macrophages. However, although authors discuss the study limitations of this pure in vitro research, the results are too preliminary and have to be expanded for publication.

Major:

Authors should detect the potential cytotoxicity of EMD in their cellular models, especially because the treatment always lowers the outcome.

Authors should increase the number of independent experiments to gain more significant results (e.g. in Figure 2, Figure 5). A number of 3 to 4 sets is in general too low for publication. In suppl. Figure 2 results from 2 experiments were shown with a p-value for statistical significance. This is not suitable!

Authors should measure expression of other pro- and anti-inflammatory mediators (e.g. TNFa, INFg, IL-6, chemokines, etc. ) in their macrophages to identify their differentiation state. Additionally, the release of ROS could be discussed in both directions because the oxidative burst is a necessary reaction especially in M2 differentiated phagocytotic macrophages. Furthermore, why are these macrophages (cell line and BMDM) have been used? Macrophages are heterogenous cells that could be divided in several subpopulations dependent on their origin and differentiation state and environment. This should be discussed in ore detail or also other macrophages subpopulations should be used.

In primary macrophages, only gene expression analysis was performed. This has to be expanded.

Minor:

Westernblot and immunofluorescence analysis in figure 4B and C has to be quantified. The described differences are not always visible.

Original blots should be labeled with sample and molecular weight.

Round 2

Reviewer 1 Report

Thank you for changing the title to reflect the data shown and removing the irrelevant figure.

Reviewer 2 Report

The authors improved the manuscript including title and mainly discussion. I agree that some of my recommendations are out of scope as the authors discussed the limitations of this in vitro study very well. The presentation of the results was modified and is now more cleat to the readers. The discussion was expanded regarding macrophage heterogeneity and differentiation.